# The spatial-temporal distribution of soil-transmitted helminth infections in Guangdong Province, China: A geostatistical analysis of data derived from the three national parasitic surveys

**Si-Yue Huang**[1☯]**, Ying-Si Lai**[1,2☯]*****, Yue-Yi Fang**[3]*****

**1** Department of Medical Statistics, School of Public Health, Sun Yat-sen University, Guangzhou, People's Republic of China, **2** Sun Yat-sen Global Health Institute, Sun Yat-sen University, Guangzhou, People's Republic of China, **3** Center for Disease Control and Prevention of Guangdong Province, Guangzhou, People's Republic of China

☯ These authors contributed equally to this work.
* laiys3@mail.sysu.edu.cn (YSL); fang-yueyi@163.com (YYF)

**Editor:** jong-Yil Chai, Seoul National University College of Medicine, REPUBLIC OF KOREA

**Data Availability Statement:** The disease data cannot be shared publicly because of the confidentiality required by Guangdong Provincial

## Abstract

### Background

The results of the latest national survey on important human parasitic diseases in 2015–2016 showed Guangdong Province is still a moderately endemic area, with the weighted prevalence of soil-transmitted helminths (STHs) higher than the national average. High-resolution age- and gender-specific spatial-temporal risk maps can support the prevention and control of STHs, but not yet available in Guangdong.

### Methodology

Georeferenced age- and gender-specific disease data of STH infections in Guangdong Province was derived from three national surveys on important human parasitic diseases, conducted in 1988–1992, 2002–2003, and 2015–2016, respectively. Potential influencing factors (e.g., environmental and socioeconomic factors) were collected from open-access databases. Bayesian geostatistical models were developed to analyze the above data, based on which, high-resolution maps depicting the STH infection risk were produced in the three survey years in Guangdong Province.

### Principal findings

There were 120, 31, 71 survey locations in the first, second, and third national survey in Guangdong, respectively. The overall population-weighted prevalence of STH infections decreased significantly over time, from 68.66% (95% Bayesian credible interval, BCI: 64.51–73.06%) in 1988–1992 to 0.97% (95% BCI: 0.69–1.49%) in 2015–2016. In 2015–2016, only low to moderate infection risk were found across Guangdong, with hookworm becoming the dominant species. Areas with relatively higher risk (>5%) were mostly

Center for Disease Control and Prevention.
Researchers can contact Guangdong Provincial
Center for Disease Control and Prevention at +86-
020-31051692 or sjkzxjfs@gd.gov.cn to apply for
the data. All other data are available from the open-
access databases.

**Funding:** YSL was financial supported by the
National Natural Science Foundation of China
(Grant No. 82073665, https://www.nsfc.gov.cn/),
and by the Natural Science Foundation of
Guangdong Province (Grant No.
2022A1515010042, http://gdstc.gd.gov.cn/), and
by the Sanming Project of Medicine in Shenzhen
(Grant No. SZSM201803061), and by the
Fundamental Research Funds for the Central
Universities, Sun Yat-sen University (Grant No.
22qntd4201). The funders had no role in study
design, data collection and analysis, decision to
publish, or preparation of the manuscript.

**Competing interests:** The authors have declared
that no competing interests exist.

distributed in the western region. Females had higher infection risk of STHs than males. The infection risk of *A. lumbricoides* and *T. trichiura* were higher in children, while middle-aged and elderly people had higher infection risk of hookworm. Precipitation, elevation, land cover, and human influence index (HII) were significantly related with STH infection risk.

## Conclusions/Significance

We produced the high-resolution, age- and gender-specific risk maps of STH infections in the three national survey periods across nearly 30 years in Guangdong Province, which can provide important information assisting the control and prevention strategies.

### Author summary

Even though the overall infection risk of soil-transmitted helminths (STHs) in Guangdong Province decreased over time, it is still higher than the national average. Risk maps can assist spatial-targeting control and intervention. We developed Bayesian geostatistical models based on the data derived from three national surveys on important human parasitic diseases in Guangdong, conducted in 1988–1992, 2002–2003, and 2015–2016, respectively. Based on these, high-resolution, age- and gender-specific infection risk maps were produced. We found that the overall STH infection risk sharply decreased over time, from 68.66% to 0.97%. We identified that moderately endemic risk (>5%) mostly distributed in small areas of western Guangdong, and prevalence in all other areas was below 5% in 2015–2016. The higher infection risk of STHs was found among females. Children were most likely to be infected with *A. lumbricoides* and *T. trichiura*, however, middle-aged and elderly people had higher infection risk of hookworm. Therefore, results of this study suggested that the government can pay more attention to people in western Guangdong, especially middle-aged and elderly people engaged in farming. As such, the government may continue to strengthen the monitoring net in the historical endemic areas to avoid the re-infection.

## Introduction

The soil-transmitted helminth (STH) infections may lead to a group of parasitic diseases including ascariasis, trichuriasis, and hookworm disease, caused by human ingesting eggs (*Ascaris lumbricoides* and *Trichuris trichiura*) or contacting with larvae (hookworm) [1,2]. These infections usually happen in low- or middle-income countries and a single person can sometimes be chronically infected with more than one type of the three parasites [3]. People with acute and chronic STH infections can have iron-deficiency anaemia and malnutrition, and children infected with STHs can have stunting, mental retardation, and cognitive deficits [1,4]. The global disease burden of STH infections in 2017 overall was 1.9 million disability-adjusted life years (DALYs), and the majority of this burden happened in sub-Saharan Africa, East Asia, South Asia, and South America [4–6].

There were three national surveys on important human parasitic diseases in China. The first one was conducted in 1988–1992, and the standardized prevalence rate (adjusted to the age and gender structures of the national population) of any STH infection was 53.21% [7,8]. Due to this result showing heavy infection risk of STH in China, the Chinese government set

up several national control plans, including "National Plan for Control of Parasitic Diseases during the Eight Five-Year", "the Control Plan for Common Helminthiasis", and "the Comprehensive Control Plan for Common Helminthiasis among Students in China", and conducted them accordingly [9,10]. Thanks to these effective control programs, the second survey conducted in 2002–2003 showed that the standardized prevalence rate of any STH infection significantly dropped to 19.34% [7,8]. Thereafter, the government established demonstration plots of integrated STHs control, and proposed a control strategy "Health education as leading, infection source control as key intervention" [9,10]. The standardized prevalence rate of any STH infection further fallen to 3.38%, according to the third national survey conducted in 2015–2016 [7,8]. Although a significant decrease since 1990, the number of people infected with STHs in China still accounted for a large proportion around the world [7,8,11].

Guangdong Province, located in the southern part of China, used to be a high endemic region of STH infections [12], with a detection STH prevalence rate (calculated as the number of positive individuals divided by the number of examined ones) of 65.06% in the first national survey [13]. In the second survey, the detection rate was declined to 16.38% [14]. The detection rate was further dropped to 2.79% in the third survey. Also, dominant species of STHs changed in the three surveys in the province. However, due to the fact that the weighted infection rate (6.37%) (calculated using the sampling weights and population structure weights of sampled individuals [15]), was still higher than the national average (4.49%), further attentions should be paid in Guangdong [16]. Results obtained from the three surveys also indicated a spatial diversity of infection risk in the province, with a relatively higher epidemic in the western areas and a lower risk in the south central areas (e.g., the Pearl River Delta) [17,18]. As a result, control strategies should be spatial-targeted, tailored to local settings. To our knowledge, researches on STH infections in Guangdong were mainly concentrated in limited number of locations [19,20], hard to reflect the infection risk at high spatial resolution across the whole province [21,22].

Risk maps depicting the spatial-temporal distribution of STH infections can provide useful information guiding control plans, assisting governments to put priority on high-risk areas, thereby increasing cost-effectiveness of control strategies [23]. Bayesian geostatistical modeling introduces spatial-temporal effects in the model and associates disease data with potential influencing factors, thus estimating the disease risk in areas without observed data [24–26]. At present, there were high-resolution risk maps depicting the distribution of STH infections in regions of Africa, America, and Asia [27–30]. Particularly, there was one study on high-resolution mapping of STH infections in China[31]. However, this study based on observed data between 2000 and 2010, without considering age- and gender-specific infection risk, thus was not able to reflect both the long-term spatial-temporal trend and current status in different genders and age groups.

In this study, we aimed to estimate the long-term spatial-temporal, age- and gender-specific infection risk of STHs in Guangdong Province, China, at high-spatial resolutions, based on Bayesian geostatistical modeling of survey data derived from the three national parasitic surveys in Guangdong and other potential climate, environmental, and socioeconomic predictors. The results can provide important information for spatial-targeting control and intervention strategies in the province.

## Methods

### Ethics statement

The survey data of soil-transmitted helminth infections in Guangdong Province was derived from the three national surveys on important human parasitic diseases, provided by

Guangdong Provincial Center for Disease Control and Prevention. The three surveys were approved by the ethics committee in the National Institute of Parasitic Diseases, Chinese Center for Disease Control and Prevention. The objectives, procedures, and potential risks were orally explained to all participants. A written consent form was also obtained from each participant, with their own signature or the signature of the proxy. All data in our study were aggregated at each gender and age groups in survey locations and did not contain information identifiable at the individual or household level.

## Study area

Guangdong Province locates in southeast China, occupying a land area of 179,725 km², from 20˚13′ to 25˚31′ north latitude and 109˚39′ to 117˚19′ east longitude (S1 Fig) [32,33]. The province has a subtropical monsoon climate, featuring warm and humid air [34], which is very suitable for the survival and reproduction of human parasites and vector hosts [35]. The province consists of 21 prefecture-level divisions, divided into four regions by the natural landscape and economic level: Pearl River Delta, East Guangdong, West Guangdong, and North Guangdong [36]. A large degree of development inequality exists between the four regions [37].

## Data source

**Disease data.**   The data on each type of STH infections (i.e., *A. lumbricoides*, *T. trichiura*, and hookworm) was derived from the three national surveys of important human parasitic diseases in Guangdong Province, presenting as number of examined and number of positive individuals in each gender (i.e., male and female) and each age group (i.e., <7 years old, 7–14 years old, 15–44 years old, 45–59 years old, and ≥60 years old) of each survey site (village or community). The Kato-Katz technique (one sample, two slide-readings) was applied as the diagnostic approach. Geographical coordinates of survey sites were obtained through Google Maps (https://www.google.com).

All surveys ensured sample representativeness by using the multiple-stage stratified cluster sampling method. According to the geographical location, landscape or natural environment, the province was divided to four sectors, Particularly, in the first national survey, Guangdong Province was divided into sectors by the geographical location (i.e., east, west, north and south) and landscape (e.g., plain, hilly region, and mountainous region); in the second survey, the province was divided according to the geographical location; and in the third survey, the province was divided based on natural environment according to the *Ecozone Classification in China* [15,38,39]. The counties in each sector were further stratified into three economic levels, resulting in 12 types of counties.

The total sampling size was calculated and distributed to the types of counties proportional to the population. The sampling unit was set as the natural village/community, each of which was sampled around 500 residents in the first and the second surveys and around 250 in the third one. The number of sampling units in each county type was calculated as the number of sampling size of the county type divided by the sample residents in each unit. Usually, each county sampled around 3 units and the number of sampled counties in each type were determined. The counties were firstly sampled among each type, and then the units in the counties were sampled. In each sampled unit, residents were further sampled [15,38,40,41]. The above sampling process was randomized.

**Environmental, socioeconomic and demographic data.**   Environmental, socioeconomic and demographic data were derived from different open-access data sources (see Table 1 for more details). We used the spatial resolution of 5×5 km² for risk mapping, and all data were linked into a raster file with resolution of 5×5 km². Land surface temperature (LST),

**Table 1. Climatic, demographic and environmental data sources[a].**

| Sources | Data Type | Time | Resolution |
|---|---|---|---|
| MODIS/Terra[b] | LST[h] at day | 2000; 2002–2003; 2015–2016 | 1 km |
| | LST[h] at night | 2000; 2002–2003; 2015–2016 | 1 km |
| | Land cover | 2001; 2002–2003; 2015 | 1 km |
| | NDVI[i] | 2000; 2002–2003; 2015–2016 | 1 km |
| Resource and Environment Science and Data Center[c] | Precipitation | 1988–1990; 2002–2003; 2015 | 1 km |
| | Humidity | 1988–1990; 2002–2003; 2015 | 1 km |
| | GDP per capita | 1995; 2005; 2015 | 1 km |
| SEDAC[d] | Urban extents | 1995 | 1 km |
| | HII[j] | 1995–2004 | 1 km |
| SRTM[e] | Elevation | 2000 | 1 km |
| NCEI[f] | Nightlight | 1992; 2002–2003; 2015–2016 | 1 km |
| WorldPop[g] | Population data | 1988–1990; 2002–2003; 2015 | 1 km |

[a]Data accessed in January 2019

[b]Moderate Resolution Imaging Spectroradiometer (MODIS)/Terra; available at: http://modis.gsfc.nasa.gov/.

[c]Resource and Environment Science and Data Center; available at: https://www.resdc.cn/.

[d]Socioeconomic Data and Applications Center (SEDAC); available at: http://sedac.ciesin.org/.

[e]Shuttle Radar Topography Mission (SRTM); available at: https://www2.jpl.nasa.gov/srtm/.

[f]National Centers for Environmental Information (NCEI); available at: https://www.ngdc.noaa.gov/ngdc.html/.

[g]The WorldPop project; available at: http://www.worldpop.org.uk/.

[h]Land surface temperature (LST).

[i]Normalized difference vegetation index (NDVI).

[j]Human influence index (HII).

normalized difference vegetation index (NDVI), precipitation, humidity, nightlight, GDP, land cover and population data in three survey periods were approximated by summarizing over the periods of the corresponding years or to the closest years. We considered the arithmetic mean and mode as the aggregated value for continuous and categorical covariates, respectively. The land cover data were reclassified from the original 17 types into the following six categories based on between-class similarities: 1) forest (including evergreen needleleaf forests, evergreen broadleaf forests, deciduous needleleaf forests, deciduous broadleaf forests, and mixed forests); 2) shrublands and grass (including closed shrublands, open shrublands, woody savannas, savannas, and grasslands); 3) wet areas (including permanent wetlands and water bodies); 4) croplands (including croplands, and cropland/natural vegetation mosaics); 5) urban (including urban and built-up lands); and 6) other types and unclassified (including permanent snow and ice, barren, and areas with null data).

## Statistical analysis

The survey time was set according to the three survey periods: 1988–1992, 2002–2003, and 2015–2016. Gender and urban extent data were regarded as binary variables. Age groups and land cover were multi-categorical variables. Continuous variables were standardized to mean 0 and standard deviation 1. To avoid collinearity, we selected one of the variables, which was more meaningful or with better data quality, among any pair of continuous variables with Pearson's correlation coefficient > 0.8. We put each categorical variable as dummy variable in the later modelling analysis.

For each STH species, Bayesian geostatistical logistic regression models with specific spatial-temporal random effects were developed, using the methodology similar to Lai et al [31],

to evaluate the geographical distribution of STH infections in Guangdong Province. We assumed that $Y_{itgk}$ followed a binomial distribution $Y_{itgk} \sim Bn(p_{itgk}, \ n_{itgk})$, and $Y_{itgk}, n_{itgk}, p_{itgk}$ represented the number of positive individuals, the number of examined people and infection prevalence of population in gender $g$ ($g = 0$ indicating male and $g = 1$ indicating female) and age groups $k$ ($k = 1, 2, 3, 4, 5$, indicating the age group of <7 years old, 7–14 years old, 15–44 years old, 45–59 years old, and ≥60 years old, respectively) at each location $i$ ($i = 1, 2, 3, \ldots, l$) and time period $t$. Then we used a linear combination of variable effects to account for the logit transformation of $p_{itgk}$ : $logit(p_{itgk}) = \beta_0 + X_{it}^T\boldsymbol{\beta} + \boldsymbol{\vartheta}_{it} + \phi_i$, where $\beta_0$ was the intercept, $X_{it}^T$ was the vector of covariates for location $i$ and survey time $t$ (age and gender groups were considered as categorical variables to develop models), and $\boldsymbol{\beta}$ stood for the vector of regression coefficients. $\boldsymbol{\vartheta}_{it}$ and $\phi_i$ were introduced to represent spatial-temporal random effect and an exchangeable non-spatial random effect. We assumed $\boldsymbol{\vartheta}_{it}$ arose from $\boldsymbol{\vartheta} \sim GP(0, Z_s \otimes Z_t)$, which was a zero-mean Gaussian distribution. The temporal covariance matrix $Z_t$ with the coefficient $|\rho| < 1$ was set to be an autoregressive process with order one (AR1). The spatial covariance matrix was a Matérn function with spatial correlation $\sigma_{sp}^2(\kappa d_{ij})^\nu K_\nu(\kappa d_{ij})/(\Gamma(\nu)2^{\nu-1})$, and the spatial variance $\sigma_{sp}^2 = 1/(4\pi\kappa^{2\nu}\tau_{sp}^2)$. $d_{ij}$ was defined as the Euclidean distance between locations $i$ and $j$, $\kappa$ denoted a scaling parameter. $K_\nu$ was the modified Bessel function of the second kind with $\nu$ fixed at 1, and $\nu$ was regarded as a smoothing parameter. The spatial range $r = \sqrt{8\nu}/\kappa$, denoting that the distance at which spatial correlation becomes negligible (<0.1). $\boldsymbol{\vartheta}$ was assumed independent of each other in different times and locations, which was

$$\text{Cov}(\vartheta_{it}, \vartheta_{jt'}) = \begin{cases} 0, if \ t \neq t' \\ \sigma_{sp}^2, if \ t = t' \end{cases}$$ . In addition, $\phi_i$ was produced from a zero-mean normal distribution $\phi_i \sim N(0, \sigma_{nonsp}^2)$. As we did not have enough prior information on the distributions of the parameters in the Bayesian models, weakly informative prior distributions were adopted as following: $\beta_0, \boldsymbol{\beta} \sim N(0, 1000)$, $\log(\tau_{sp}) \sim lognormal(1, 100)$, $\log(\kappa) \sim lognormal(1,100)$, $1/\sigma_{nonsp}^2 \sim gamma(1, 0.00005)$, and $\log((1+\rho)/(1-\rho)) \sim N(0, 0.15)$, to avoid subjectively selecting unreasonable informative priors.

## Variable selection

To identify the best set of fixed effect covariates, we applied Bayesian variable selection. Firstly, to capture the potential non-linear correlations of continuous predictor with the infection risk, we considered two forms (i.e., continuous and categorical) for the continuous predictors in the model. They were constructed as three-level categorical variables based on preliminary exploratory graphical analysis. For each continuous predictors, two univariate Bayesian geostatistical models were developed, with the predictor as the only fixed effect independent variable in a continuous or categorical form in either model. We then chose the best functional form as the one with the smallest log score among the two models. Secondly, the backward elimination approach was adopted for selecting the best variables set to develop the final models [42]. According to previous research, gender and age were important factors for STH infections, and were kept in the models during variable selection [14,15,43].

## Model validation

We adopted model validation, which used a random sample 80% of observed locations as the training set, and the remaining 20% as the validating set. The training set was applied to fit the model, from which parameter estimates were obtained and were further used to estimate the age- and gender specific prevalence ($\hat{\pi}_{igk}$) at locations of the validating set. We compared the

observed ($\pi_{igk}$) and estimated prevalence ($\hat{\pi}_{igk}$) at each observation of the validation set (with the total number of observations $N$), by calculating the mean error ($ME = \frac{1}{N} \times \sum_{i=1} \sum_{g=1} \sum_{k=1} (\pi_{igk} - \hat{\pi}_{igk})$), the mean absolute error ($MAE = \frac{1}{N} \times \sum_{i=1} \sum_{g=1} \sum_{k=1} |\pi_{igk} - \hat{\pi}_{igk}|$), the coverage of observations within 95% Bayesian credible intervals (BCIs) of posterior distribution of estimated prevalence, and the area under the receiver operating characteristic (ROC) curve (AUC) to evaluate the performance of the model.

### High-resolution maps

The risk of each species of STH infections was estimated over a grid of 9,599 pixels across Guangdong Province at 5×5 km² spatial resolution. Assuming that the infectious risk of any two species of STHs was independent, we applied the following formula to estimate the infection rate of any STH species:

$$p_S = 1 - (1 - p_A) \times (1 - p_T) \times (1 - p_h),$$

where $p_S$, $p_A$, $p_T$ and $p_h$ respectively represent the estimated infection prevalence of any STH, *A. lumbricoides*, *T. trichiura*, and hookworm [30,31]. We also produced maps of the probability that prevalence exceeds a certain threshold for any STH infection. The prevalence thresholds for preventive chemotherapy (20%) [44] and transmission control (1%) were used [45].

All statistical analyses were carried out by R software (version 4.0.3), particularly with INLA R-package for parameter estimation [46]. We shared the corresponding R-codes in GitHub (https://github.com/SYSU-STH/Spatial-temporal-mapping-of-STH). The high-resolution maps of disease risk were produced through ArcGIS 10.2.

## Results

### Data summaries

There were 120, 31, 71 survey locations, with a total of 61517, 17013, 12401 individuals in the first, second and third national survey in Guangdong Province, respectively (Fig 1). Table 2 lists an overview of the characteristic of the survey data.

### Bayesian model fitting and variable selection

The selected variables were shown in Table 3. For all three species, we found that the infection risk decreased significantly over time, and females had a higher risk than males. Moreover, the difference in infection prevalence was significant between age groups. The infection risk of *A. lumbricoides* and *T. trichiura* were highest in age group of 7–14 years, then decreased following the increase of age. Hookworm infection risk was higher in the order age groups, with the age group of 45–59 years the highest, and lower in the younger age groups. For *A. lumbricoides*, elevation and precipitation were positively correlated with infection risk, while the urban areas had lower risk than other types of land cover. For *T. trichiura*, precipitation was positively correlated with infection prevalence. For hookworm, elevation had a positive relationship with infection risk, while HII showed a negative correlation.

### Model validation

Model validation suggested that the final Bayesian geostatistical models had good performance (S1 Text). They were able to correctly estimate (within the 95% BCIs) 82.3%, 70.9%, 71.8%, and 81.0% of locations for *A. lumbricoides*, *T. trichiura*, hookworm, and any STH infection,

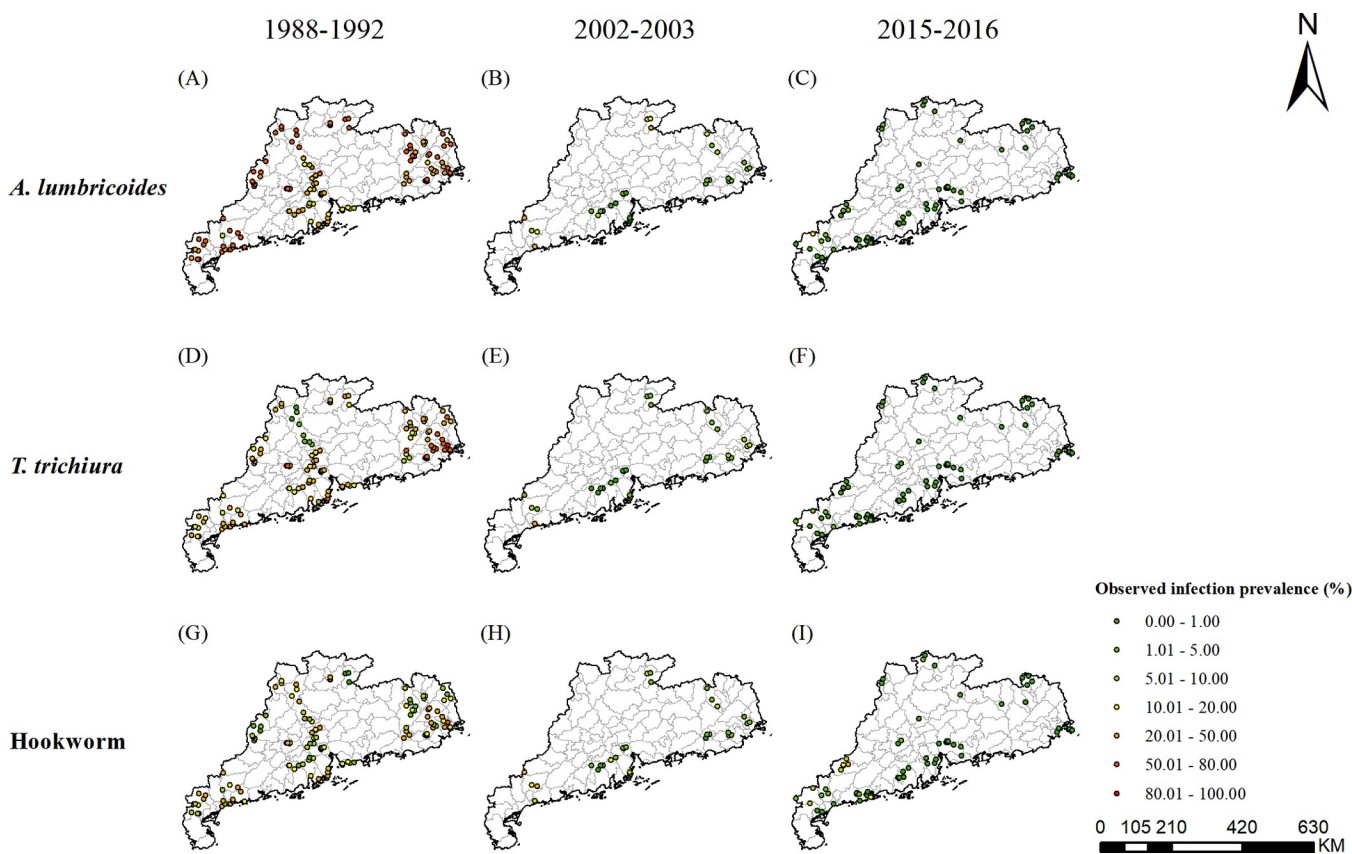

**Fig 1. Survey locations and observed prevalence in Guangdong.** (A)-(C) for *A. lumbricoides*; (D)-(F) for *T. trichiura*; (G)-(I) for hookworm, during the periods of 1988–1992, 2002–2003 and 2015–2016, respectively. The base layer derived from https://www.webmap.cn/mapDataAction.do?method=forw&keysearch=indexSearch with credit to National Catalogue Service For Geographic Information.

respectively. The AUCs were 0.95, 0.94, 0.88, and 0.95 for *A. lumbricoides*, *T. trichiura*, hookworm, and any STH infection, respectively, suggesting good estimated capacities [47].

## Estimated infection prevalence and number of people infected

Show in Table 4, the population-weighted prevalence of any STH was highest in 1988–1992 (68.66%, 95%BCI: 64.51%-73.06%). In 2002–2003, The prevalence decreased to 14.35% (95% BCI: 11.04%-20.52%), and in 2015–2016, it was further decreased to 0.97% (95% BCI: 0.69%-1.49%). The estimated number of infected individuals of any STH decreased from 43.13 million (95% BCI: 40.53–45.90) in 1988–1992 to 11.09 million (95% BCI: 8.52–15.85) in 2002–2003, and further to 1.05 million (95% BCI: 0.74–1.62) in 2015–2016. Moreover, dominant species of STHs in 1988–1992 was *A. lumbricoides*, with estimated prevalence of 41.03% (95% BCI: 37.23%-46.00%), while in 2002–2003 and 2015–2016, the dominant species were hookworm, with estimated prevalence of 7.59% (95% BCI: 4.50%-13.88%) and 0.61% (95% BCI: 0.36%-1.11%), respectively.

## Estimated risk maps

The risk maps of *A. lumbricoides*, *T. trichiura*, hookworm, and any STH in different survey periods were presented in Figs 2–5, respectively, which showed obvious decreased trends over time in Guangdong Province. For *A. lumbricoides*, the estimated infection risk was generally

**Table 2. Overview of characteristic of STH infection surveys data in Guangdong Province.**

| Survey year | 1988–1992 | 2002–2003 | 2015–2016 | Total |
|---|---|---|---|---|
| Survey location | 120 | 31 | 71 | 205[a] |
| Number of individuals | 61517 | 17013 | 12401 | 90931 |
| Gender | | | | |
| Male | 29004 | 8422 | 6165 | 43591 |
| Female | 32513 | 8591 | 6236 | 47340 |
| Age (years) | | | | |
| < 7 | 10571 | 2045 | 1481 | 14097 |
| 7–14 | 12024 | 3836 | 1944 | 17804 |
| 15–44 | 27118 | 6853 | 4478 | 38449 |
| 45–59 | 7112 | 2765 | 2600 | 12477 |
| ≥ 60 | 4692 | 1514 | 1898 | 8104 |
| Raw prevalence (%) | | | | |
| *A. lumbricoides* | 46.44 | 7.80 | 0.52 | 32.95 |
| *T. trichiura* | 33.19 | 5.70 | 0.46 | 23.58 |
| Hookworm | 15.84 | 5.77 | 1.88 | 12.05 |
| Any STH | 65.06 | 16.38 | 2.79 | 47.46 |

[a]Locations after dropping the duplicated ones

high (>20%) in the study region in 1988–1992, particularly in the northern and the western part, with the estimated prevalence higher than 50%. In period 2002–2003, the estimated infection risk dropped sharply. Low prevalence (<5%) were estimated for Pearl River Delta and the eastern part of Guangdong, and moderate prevalence (5–20%) was distributed in large areas of northern and western Guangdong. However, in several areas of northern and western, high infection prevalence (>20%) was remained. In 2015–2016, the prevalence in most areas of Guangdong was less than 1%, while in a few areas of western Guangdong, it was 1% to 5%. For *T. trichiura*, in 1988–1992, except for areas of the northern part of Guangdong, high prevalence (>20%) was estimated in large areas, particularly in cities of Chaozhou, Shantou, Meizhou, Jieyang, with the estimated prevalence more than 50%. In 2002–2003, low prevalence (<5%) was presented in most areas, and only a few small areas in the western part showed moderate prevalence (5–20%). In 2015–2016, very low prevalence (<1%) was estimated across the whole province. For hookworm, in 1988–1992, areas of northern, western, and eastern Guangdong showed high prevalence (>20%). In 2002–2003, infection of hookworm was just prevalent in several areas of cities of Maoming, Shaoguan, and Meizhou in the western part, with moderate prevalence (5–20%). In 2015–2016, except for small parts of western Guangdong with moderate prevalence (5–10%), most areas in the province presented low prevalence (<5%).

In terms of infection of any STH, in 1988–1992, most areas of the whole province show very high prevalence (>50%). From the probability contour map (Fig 6A), we found that most areas in the whole province were with very high probabilities that the prevalence exceeds 20%. In 2002–2003, The infection was still serious in the western and the northern parts, with high prevalence (>20%) in very high probabilities. Only in the Pearl River Delta, the prevalence was less than 10%. In 2015–2016, infection risk was mainly concentrated in small areas of the western, with moderate prevalence (5–10%), and prevalence in all other areas was below 5%. As shown in the probability contour map (Fig 6F), Pearl River Delta and the eastern parts with low probabilities (<5%) that the prevalence exceeds the transmission control threshold of 1%,

**Table 3. Posterior summaries of the geostatistical model parameters for *A. lumbricoides*, *T. trichiura*, hookworm.**

| Variable | Estimated median (95% BCI) | | |
|---|---|---|---|
| | *A. lumbricoides* | *T. trichiura* | Hookworm |
| Year | | | |
| 2015–2016 | Ref | Ref | Ref |
| 2002–2003 | 4.30 (3.19, 5.45)[a] | 2.51 (1.33, 3.71)[a] | 3.16 (0.87, 5.45)[a] |
| 1988–1992 | 7.19 (5.97, 8.44)[a] | 5.67 (4.39, 6.96)[a] | 4.71 (2.81, 6.63)[a] |
| Gender | | | |
| Male | Ref | Ref | Ref |
| Female | 0.17 (0.14, 0.20)[a] | 0.07 (0.03, 0.11)[a] | 0.24 (0.20, 0.29)[a] |
| Age (years) | | | |
| <7 | 0.11 (0.06, 0.16)[a] | -0.57 (-0.63, -0.51)[a] | -1.87 (-1.97, -1.78)[a] |
| 7–14 | 0.36 (0.31, 0.40)[a] | 0.59 (0.54, 0.64)[a] | -1.13 (-1.20, -1.06)[a] |
| 15–44 | Ref | Ref | Ref |
| 45–59 | -0.09 (-0.15, -0.04)[a] | -0.10 (-0.16, -0.03)[a] | 0.46 (0.40, 0.52)[a] |
| ≥60 | -0.24 (-0.30, -0.17)[a] | -0.20 (-0.27, -0.12)[a] | 0.30 (0.22, 0.37)[a] |
| Precipitation (mm) | | | — |
| <1641.77 | Ref | Ref | |
| 1641.77–1725.32 | 0.42 (-0.01,0.85) | 0.26 (-0.23, 0.74) | |
| ≥1725.32 | 0.58 (0.02,1.14)[a] | 0.81 (0.15, 1.47)[a] | |
| Land cover | | — | — |
| Shrublands and grass | Ref | | |
| Forest | -0.59 (-2.04, 0.81) | | |
| Wet areas | 0.20 (-1.02, 1.40) | | |
| Croplands | 0.25 (-0.30, 0.79) | | |
| Urban | -0.68 (-1.09, -0.28)[a] | | |
| Other types and unclassified | -1.08 (-1.97, -0.19)[a] | | |
| Elevation | 0.30 (0.02, 0.57)[a] | — | |
| <16.23 | | | Ref |
| 16.23–101.50 | | | 0.75 (0.23, 1.28)[a] |
| ≥101.50 | | | 0.60 (-0.13, 1.34) |
| HII | — | — | -0.28 (-0.53, -0.04)[a] |
| Range (km) | 211.00 (93.17, 540.09) | 158.40 (95.16, 265.19) | 160.79 (94.29, 273.80) |
| Spatial variance | 1.12 (0.52, 2.49) | 1.93 (1.05, 3.59) | 3.11 (1.56, 6.22) |
| Non-spatial variance | 0.48 (0.33, 0.70) | 0.42 (0.29, 0.62) | 0.82 (0.58, 1.15) |
| Autoregressive coefficient in AR1 | 0.60 (0.12, 0.84) | 0.59 (0.16, 0.81) | -0.25 (-0.65, 0.27) |

[a]Statistical significance

while the western Guangdong was difficult to achieve transmission control. These above results suggest that the STHs epidemic areas gradually shrank from the whole province to the western Guangdong over time. The gender- and age-specific high-resolution risk maps were shown in S2–S10 Figs.

## Discussion

In this study, based on observed infection prevalence of STHs from the three national surveys in Guangdong Province, we applied Bayesian geostatistical analysis to estimate the high-resolution risk profiles of STH infections. Particularly, age- and gender-specific risk maps for all species and any STH were produced. Besides, areas historically or currently endemic with

**Table 4. Population-adjusted estimated prevalence (%) and number of individuals (×10⁶) infected with STHs in Guangdong Province.**

|  | Estimated prevalence (%) | Estimated No. of people infected (×10⁶) |
|---|---|---|
| *A. lumbricoides* |  |  |
| 1988–1992 | 41.03 (37.23, 46.00) | 25.77 (23.39, 28.90) |
| 2002–2003 | 5.29 (3.79, 7.72) | 4.08 (2.93, 5.96) |
| 2015–2016 | 0.12 (0.07, 0.22) | 0.13 (0.07, 0.24) |
| *T. trichiura* |  |  |
| 1988–1992 | 34.84(29.58, 40.38) | 21.89 (18.58, 25.37) |
| 2002–2003 | 2.40(1.64, 3.91) | 1.86 (1.27, 3.02) |
| 2015–2016 | 0.24(0.14, 0.41) | 0.26 (0.15, 0.45) |
| Hookworm |  |  |
| 1988–1992 | 20.24 (16.05, 25.97) | 12.72 (10.08, 16.31) |
| 2002–2003 | 7.59 (4.50, 13.88) | 5.86 (3.48, 10.72) |
| 2015–2016 | 0.61 (0.36, 1.11) | 0.66 (0.39, 1.21) |
| Any STH |  |  |
| 1988–1992 | 68.66 (64.51, 73.06) | 43.13 (40.53, 45.90) |
| 2002–2003 | 14.35 (11.04, 20.52) | 11.09 (8.52, 15.85) |
| 2015–2016 | 0.97 (0.69, 1.49) | 1.05 (0.74, 1.62) |

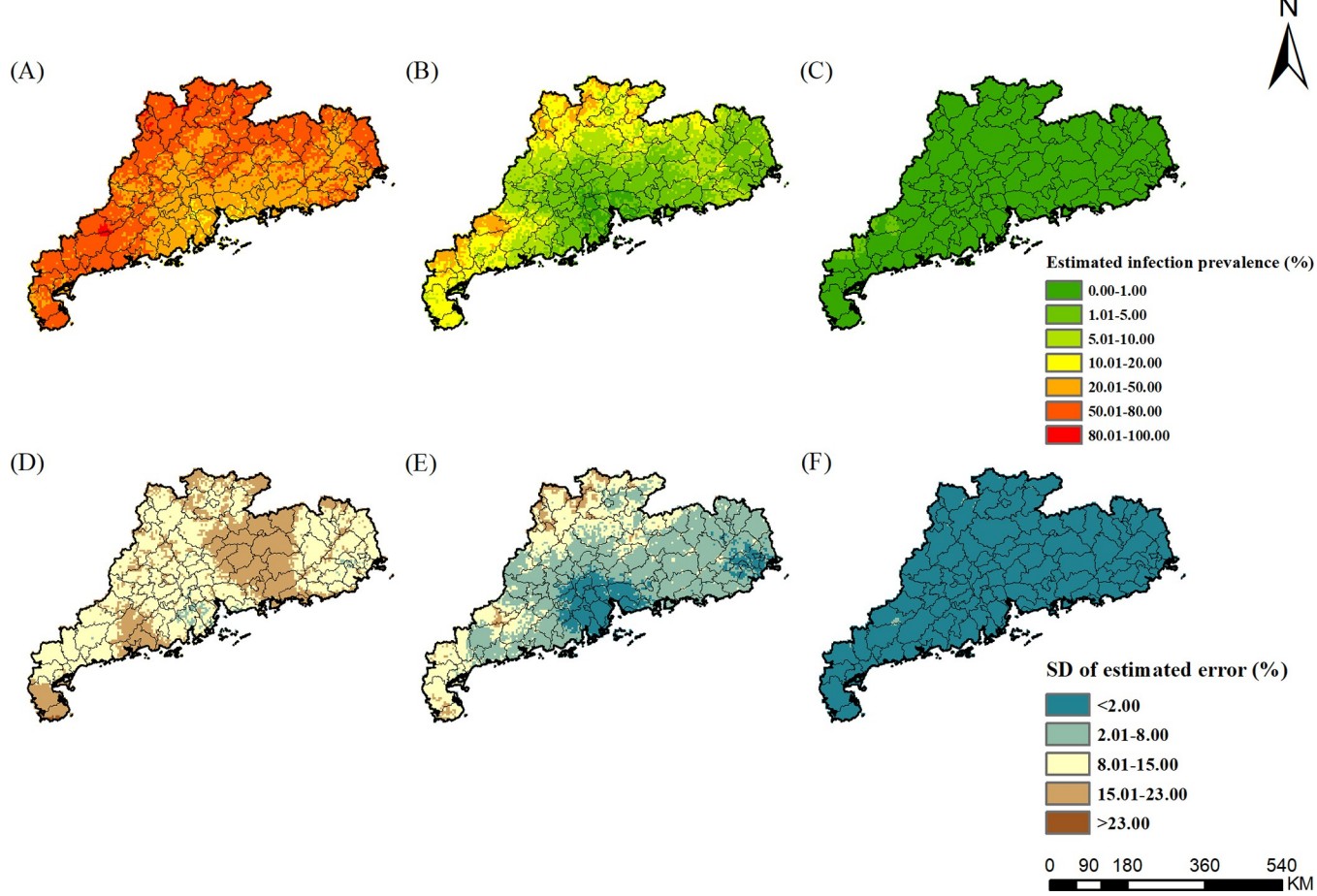

**Fig 2. The geographical distribution of *A. lumbricoide* infection risk in Guangdong.** (A)-(C) present *A. lumbricoide* infection during the periods of 1988–1992, 2002–2003, and 2015–2016, respectively. (D)-(F) present the corresponding standard deviation of the estimated uncertainty. The base layer derived from https://www.webmap.cn/mapDataAction.do?method=forw&keysearch=indexSearch with credit to National Catalogue Service For Geographic Information.

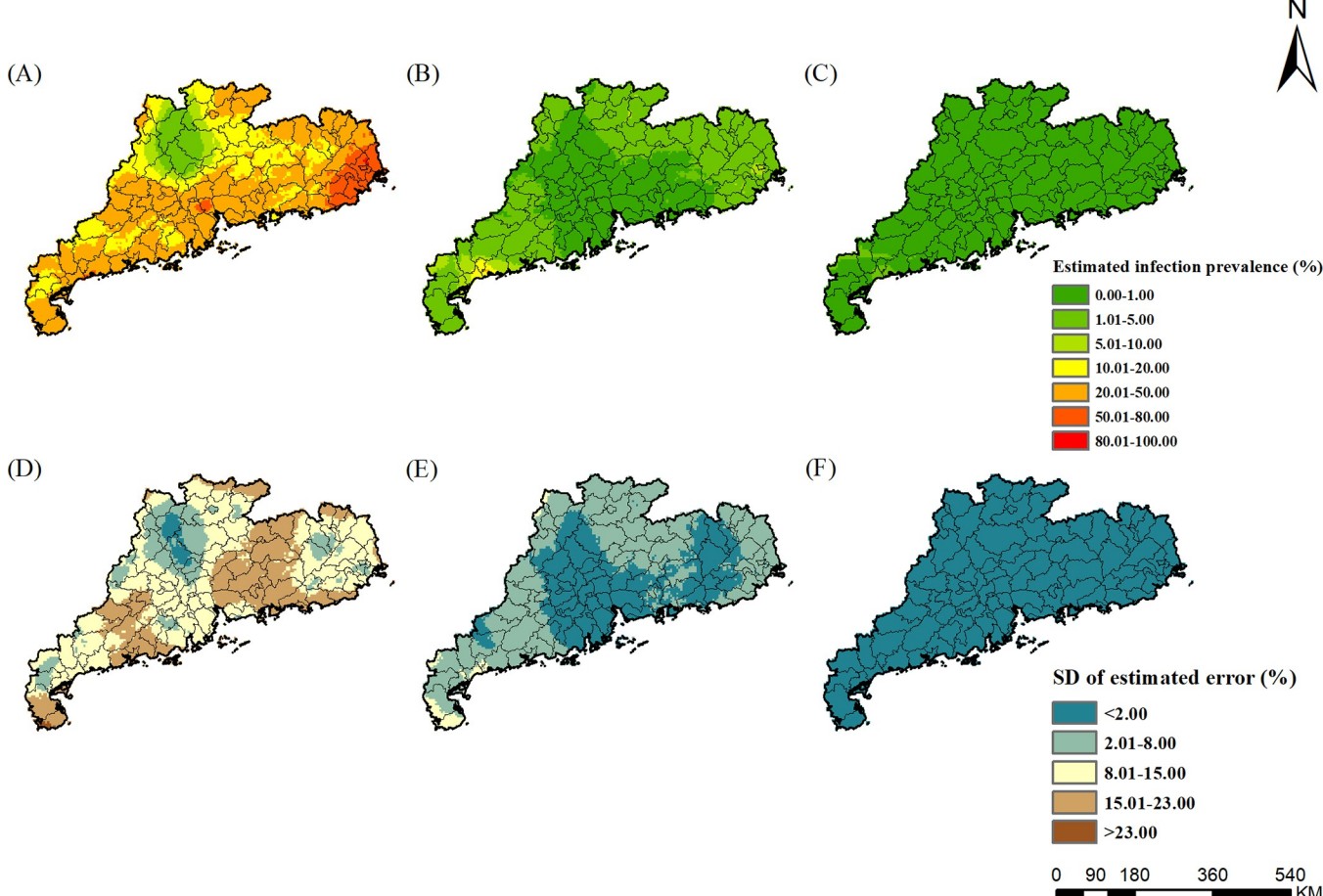

**Fig 3. The geographical distribution of *T. trichiura* infection risk in Guangdong.** (A)-(C) present *T. trichiura* infection during the periods of 1988–1992, 2002–2003, and 2015–2016, respectively. (D)-(F) present the corresponding standard deviation of the estimated uncertainty. The base layer derived from https://www.webmap.cn/mapDataAction.do?method=forw&keysearch=indexSearch with credit to National Catalogue Service For Geographic Information.

STHs were identified. Our results showed that even though the overall population-weighted prevalence of any STH infection was very lower (less than 1%) in the latest survey period, a few areas in the western regions of Guangdong were still in moderately endemic status.

We found that the infection risk of the three species of STHs dropped obviously over time, which may be attributed to a series of STHs control programs in Guangdong [17]. After the first national survey, the former Ministry of Health and the former National Education Commission launched the "National Control Program for Student's Helminth Infection" [9], which Guangdong implemented well. Besides, began from Kaiping city in 1993, the government implemented the two-stage mass drug administration (MDA) in epidemic areas, with the first stage targeting mostly students from elementary and middle schools, and the second stage on the whole residents [48,49]. In 2006, right after the second national survey, the former Ministry of Health, launched the "National Program for Control of Major Parasitic Diseases during 2006–2015" [9]. Meanwhile, in order to promote the program, a national surveillance network consisting 22 national surveillance spots on STHs was established in China, 7 of which were in Guangdong [9]. In addition, the considerable reduction in prevalence was also attributable to the rapid development of economy, urbanization in rural areas, improvement of education,

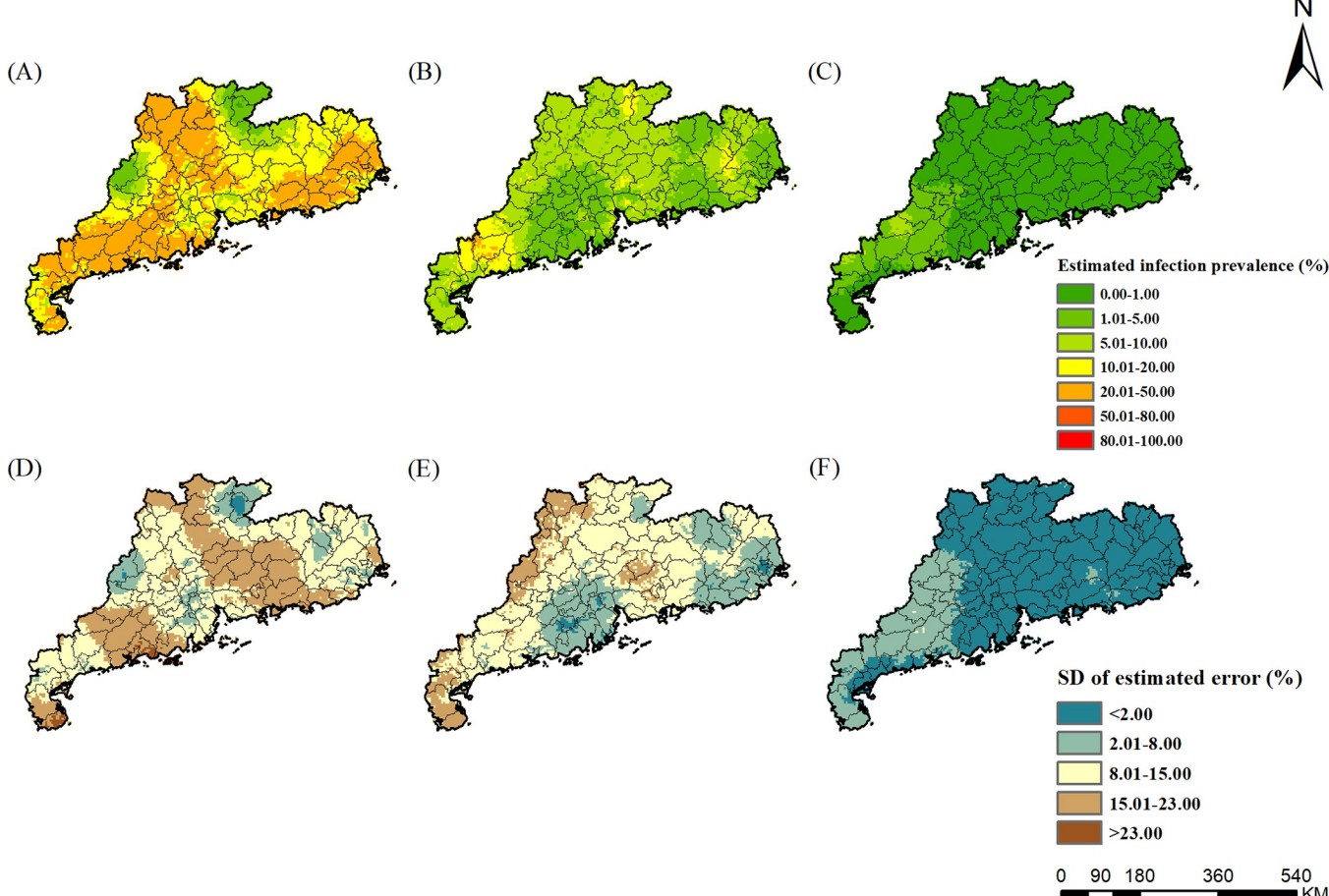

**Fig 4. The geographical distribution of hookworm infection risk in Guangdong.** (A)-(C) present hookworm infection during the periods of 1988–1992, 2002–2003, and 2015–2016, respectively. (D)-(F) present the corresponding standard deviation of the estimated uncertainty. The base layer derived from https://www.webmap.cn/mapDataAction.do?method=forw&keysearch=indexSearch with credit to National Catalogue Service For Geographic Information.

improvement of personal hygiene habits, and increased access to tap water and sanitary toilets [50,51].

Despite a sharp decrease of STH infections over time in Guangdong Province, uneven spatial distribution existed. Particularly, some areas in the western part were with relatively higher hookworm infection risk in the latest survey period. This was most likely due to the fact that the climatic condition in western part is quite suitable for the survival and reproduction of STHs [43]. Besides, large areas in the western are for farming, providing more opportunities for local people to contact with soil [52]. Meanwhile, lower economic level and the lack of public hygiene awareness of local people in the western part may increase the difficulty of delivering control interventions [53]. Moreover, the dominant species of STHs were found altered from *A. lumbricoides* to hookworm, similar to many other provinces [54–58]. One of the reasons may be the drug resistance of hookworm. At present, albendazole and mebendazole are the two most widely used drugs for preventive chemotherapy [59]. However, previous study shows that the cure rate of these drugs for hookworm was lower than for *A. lumbricoides* and *T. trichiura* [59,60]. Besides, as the infection of hookworm are through human contacting with larvae, different with the infection way of the other two species, the improvement sanitation

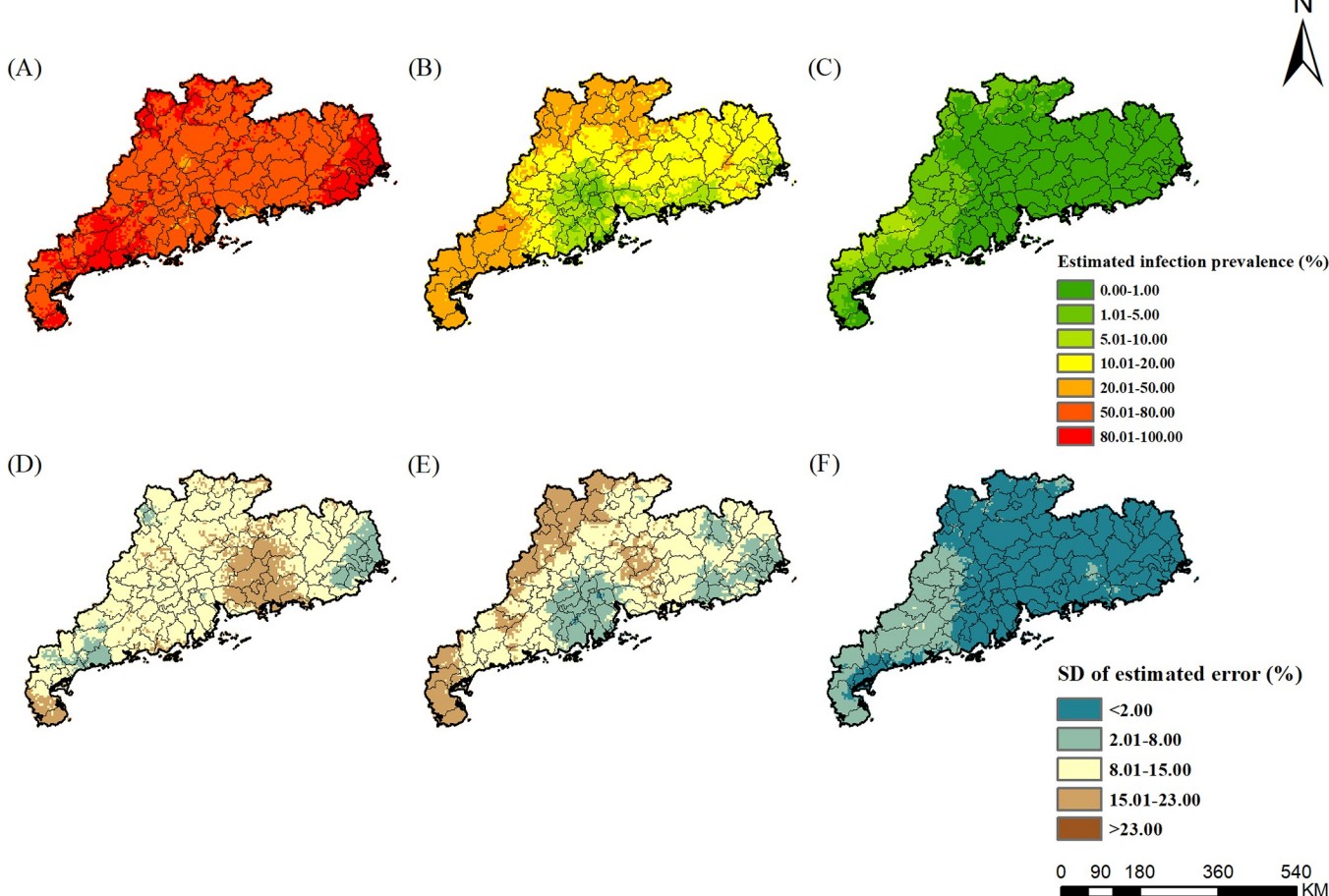

**Fig 5. The geographical distribution of any STH infection risk in Guangdong.** (A)-(C) present any STH infection during the periods of 1988–1992, 2002–2003, and 2015–2016, respectively. (D)-(F) present the corresponding standard deviation of the estimated uncertainty. The base layer derived from https://www.webmap.cn/mapDataAction.do?method=forw&keysearch=indexSearch with credit to National Catalogue Service For Geographic Information.

and drinking water in current years seems more effective for *A. lumbricoides* and *T. trichiura* than hookworm [53].

Our results shows that women had a higher infection risk of STHs than men. The possible reason was that women were the main labor force in the field and barefooted farming increased their risk to infection [17]. We also found that children aged 7 to 14 were with higher risk of *A. lumbricoides* and *T. trichiura*, as children usually lacked hygiene awareness [17,61]. While hookworm infection was more likely happened in adults, especially people aged 45 and older, which may due to that adults have more chance to be infected through agricultural activities[57]. This age distribution was similar as other previous studies [15,62]. In addition, we found several factors that were significantly associated with STH infections in Guangdong, such as elevation, precipitation, land cover, and HII. For example, our results suggested that elevation was a risk factor for *T. trichiura* and hookworm, consistent with prior studies [63,64]. One possible reason was that the higher altitude areas were mostly mountains and hills, where the economic development was inferior, and the sanitary conditions were poorer. Precipitation showed a positive relationship with *A lumbricoides* and *T. trichiura* infections. High precipitation may provide a suitable moist environment for eggs and larvae to grow [65,66]. In addition, we identified urban and areas with higher HII may lower hookworm

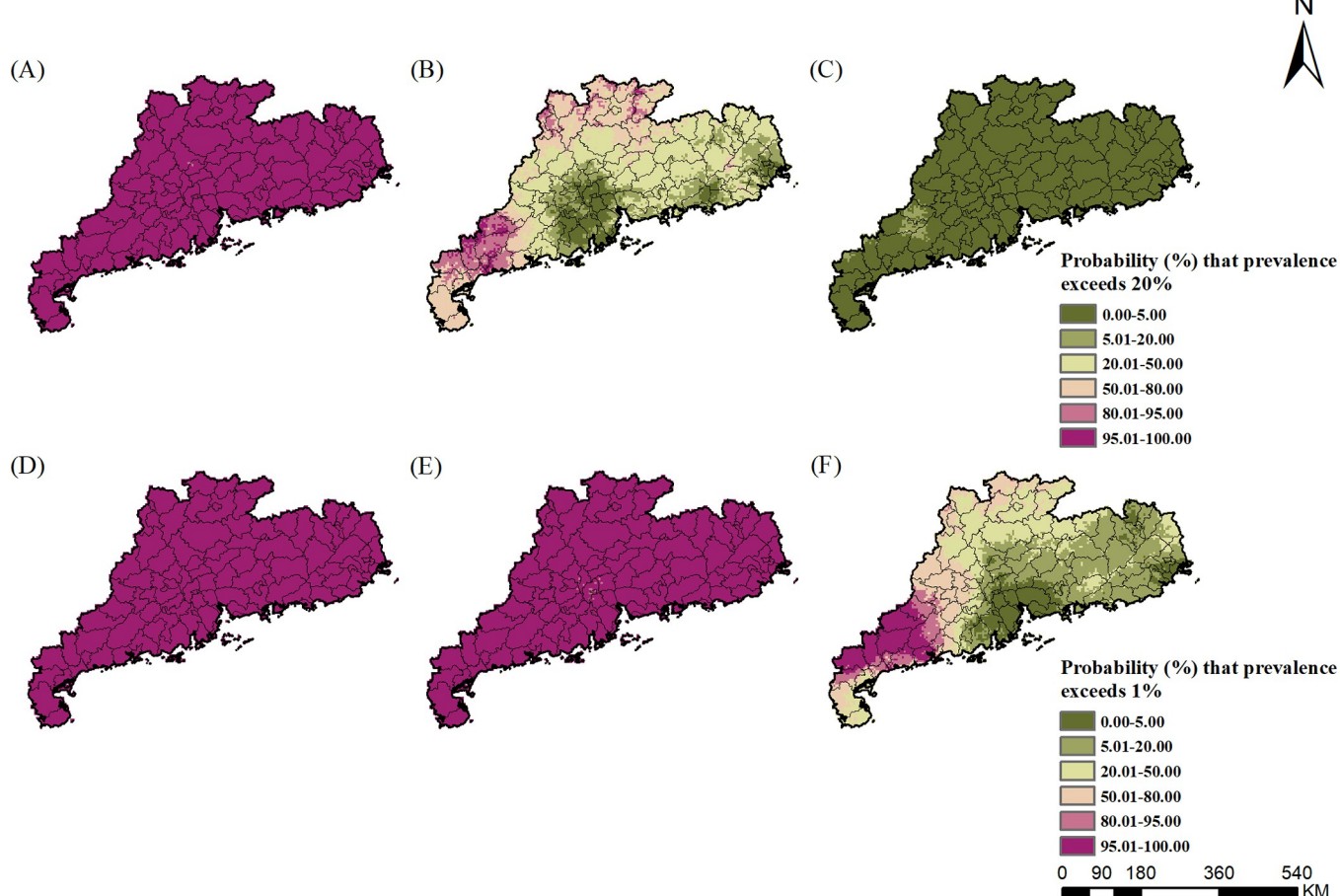

**Fig 6. Probability contour maps of any STH infection risk in Guangdong.** (A)-(C) present the probability that prevalence exceeds the threshold for preventive chemotherapy (20%); (D)-(F) present the probability that prevalence exceeds the threshold of transmission control (1%), during the periods of 1988–1992, 2002–2003, and 2015–2016, respectively. The base layer derived from https://www.webmap.cn/mapDataAction.do?method=forw&keysearch= indexSearch with credit to National Catalogue Service For Geographic Information.

infection risk, which suggested higher economic level may be a protective factor for hookworm infection [66].

Even though the overall infection risk of all the three species of STHs were in a low endemic level, further control activities should be considered in western Guangdong, particularly in areas with moderate prevalence. On one hand, children, women, middle-aged and elderly people engaged in farming should be prioritized for monitoring and provided more education opportunities [9,67]. On the other hand, due to that western Guangdong is a region with less developed economy, the local government may combine poverty alleviation programs with precise control measures, according to actual situations [7,9]. As for the historical endemic areas, the government may continue to strengthen the monitoring net, thus avoiding the re-infection [68,69]. Besides, in order to brace for potential drug resistance, the government may provide financial support to development a new field-applicable diagnostics that can effectively identify and monitor signs of emerging resistance [70].

Frankly, some limitations should be aware in our study. First, due to the absence of occupation and educational information, we didn't consider these two factors as potential predictors, which were found related to STH infections in previous studies [14,16,20,71]. Nevertheless, we considered the socioeconomic factors such as urban extent, nightlight, HII in variable selection

procedure, partially taking into account the effects of the two factors. Second, due to the difference of sampling and diagnostic methods in other surveys on STHs in Guangdong, we only used data derived from the three national surveys of important human parasitic diseases, rather than considering available data from other surveys, to maintain data consistency. Nevertheless, the three national survey data had comprehensive coverage and good representativeness. In addition, the model validation showed reasonable capacities of estimation accuracy. Third, because the latest national survey was conducted in 2014–2016, we estimated the STH infection risk till then. If further representative and comparable surveys are conducted, we should update our results.

## Conclusion

In conclusion, we present the high-resolution, age- and gender-specific risk maps of STH infections in Guangdong Province, in the three national survey periods across nearly 30 years, which provides valuable information to assist control and prevention strategies for the province.

## Supporting information

**S1 Fig. Location of Guangdong province and its 21 prefecture-level divisions.** The base layer derived from https://www.webmap.cn/mapDataAction.do?method=forw&keysearch= indexSearch with credit to National Catalogue Service For Geographic Information. (TIF)

**S2 Fig. The geographical distribution of *A. lumbricoide* infection risk in Guangdong, 1988–1990.** (A)-(E) present *A. lumbricoide* infection of males 0 to 6, 7 to 14, 15 to 44, 45 to 59, and 60 years old and older, (F)-(J) present *A. lumbricoide* infection of females 0 to 6, 7 to 14, 15 to 44, 45 to 59, and 60 years old and older, respectively. The base layer derived from https:// www.webmap.cn/mapDataAction.do?method=forw&keysearch=indexSearch with credit to National Catalogue Service For Geographic Information. (TIF)

**S3 Fig. The geographical distribution of *A. lumbricoide* infection risk in Guangdong, 2002–2003.** (A)-(E) present *A. lumbricoide* infection of males 0 to 6, 7 to 14, 15 to 44, 45 to 59, and 60 years old and older, (F)-(J) present *A. lumbricoide* infection of females 0 to 6, 7 to 14, 15 to 44, 45 to 59, and 60 years old and older, respectively. The base layer derived from https:// www.webmap.cn/mapDataAction.do?method=forw&keysearch=indexSearch with credit to National Catalogue Service For Geographic Information. (TIF)

**S4 Fig. The geographical distribution of *A. lumbricoide* infection risk in Guangdong, 2015–2016.** (A)-(E) present *A. lumbricoide* infection of males 0 to 6, 7 to 14, 15 to 44, 45 to 59, and 60 years old and older, (F)-(J) present *A. lumbricoide* infection of females 0 to 6, 7 to 14, 15 to 44, 45 to 59, and 60 years old and older, respectively. The base layer derived from https:// www.webmap.cn/mapDataAction.do?method=forw&keysearch=indexSearch with credit to National Catalogue Service For Geographic Information. (TIF)

**S5 Fig. The geographical distribution of *T. trichiura* infection risk in Guangdong, 1988– 1992.** (A)-(E) present *T. trichiura* infection of males 0 to 6, 7 to 14, 15 to 44, 45 to 59, and 60 years old and older, (F)-(J) present *T. trichiura* infection of females 0 to 6, 7 to 14, 15 to 44, 45 to 59, and 60 years old and older, respectively. The base layer derived from https://www.

webmap.cn/mapDataAction.do?method=forw&keysearch=indexSearch with credit to National Catalogue Service For Geographic Information.
(TIF)

**S6 Fig. The geographical distribution of *T. trichiura* infection risk in Guangdong, 2002–2003.** (A)-(E) present *T. trichiura* infection of males 0 to 6, 7 to 14, 15 to 44, 45 to 59, and 60 years old and older, (F)-(J) present *T. trichiura* infection of females 0 to 6, 7 to 14, 15 to 44, 45 to 59, and 60 years old and older, respectively. The base layer derived from https://www.webmap.cn/mapDataAction.do?method=forw&keysearch=indexSearch with credit to National Catalogue Service For Geographic Information.
(TIF)

**S7 Fig. The geographical distribution of *T. trichiura* infection risk in Guangdong, 2015–2016.** (A)-(E) present *T. trichiura* infection of males 0 to 6, 7 to 14, 15 to 44, 45 to 59, and 60 years old and older, (F)-(J) present *T. trichiura* infection of females 0 to 6, 7 to 14, 15 to 44, 45 to 59, and 60 years old and older, respectively. The base layer derived from https://www.webmap.cn/mapDataAction.do?method=forw&keysearch=indexSearch with credit to National Catalogue Service For Geographic Information.
(TIF)

**S8 Fig. The geographical distribution of hookworm infection risk in Guangdong, 1988–1992.** (A)-(E) present hookworm infection of males 0 to 6, 7 to 14, 15 to 44, 45 to 59, and 60 years old and older, (F)-(J) present hookworm infection of females 0 to 6, 7 to 14, 15 to 44, 45 to 59, and 60 years old and older, respectively. The base layer derived from https://www.webmap.cn/mapDataAction.do?method=forw&keysearch=indexSearch with credit to National Catalogue Service For Geographic Information.
(TIF)

**S9 Fig. The geographical distribution of hookworm infection risk in Guangdong, 2002–2003.** (A)-(E) present hookworm infection of males 0 to 6, 7 to 14, 15 to 44, 45 to 59, and 60 years old and older, (F)-(J) present hookworm infection of females 0 to 6, 7 to 14, 15 to 44, 45 to 59, and 60 years old and older, respectively. The base layer derived from https://www.webmap.cn/mapDataAction.do?method=forw&keysearch=indexSearch with credit to National Catalogue Service For Geographic Information.
(TIF)

**S10 Fig. The geographical distribution of hookworm infection risk in Guangdong, 2015–2016.** (A)-(E) present hookworm infection of males 0 to 6, 7 to 14, 15 to 44, 45 to 59, and 60 years old and older, (F)-(J) present hookworm infection of females 0 to 6, 7 to 14, 15 to 44, 45 to 59, and 60 years old and older, respectively. The base layer derived from https://www.webmap.cn/mapDataAction.do?method=forw&keysearch=indexSearch with credit to National Catalogue Service For Geographic Information.
(TIF)

**S1 Text. The results of model validation.**
(DOCX)

## Acknowledgments

We are grateful to Ye Liu from Sun Yat-sen University, Guangzhou, China for the help in data collection and analysis.

## Author Contributions

**Conceptualization:** Ying-Si Lai, Yue-Yi Fang.

**Data curation:** Si-Yue Huang, Yue-Yi Fang.

**Formal analysis:** Si-Yue Huang, Ying-Si Lai.

**Funding acquisition:** Ying-Si Lai.

**Methodology:** Si-Yue Huang, Ying-Si Lai, Yue-Yi Fang.

**Project administration:** Si-Yue Huang.

**Supervision:** Ying-Si Lai.

**Validation:** Si-Yue Huang, Ying-Si Lai, Yue-Yi Fang.

**Visualization:** Si-Yue Huang, Ying-Si Lai.

**Writing – original draft:** Si-Yue Huang, Ying-Si Lai.

**Writing – review & editing:** Si-Yue Huang, Ying-Si Lai, Yue-Yi Fang.

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
