## [Decision Letter · Decision Letter 0]

25 Apr 2022

Dear Dr. Lai,

Thank you very much for submitting your manuscript "The spatial-temporal distribution of soil-transmitted helminth infections in Guangdong Province, China: a geostatistical analysis of data derived from the three national parasitic surveys" for consideration at PLOS Neglected Tropical Diseases. As with all papers reviewed by the journal, your manuscript was reviewed by members of the editorial board and by several independent reviewers. The reviewers appreciated the attention to an important topic. Based on the reviews, we are likely to accept this manuscript for publication, providing that you modify the manuscript according to the review recommendations. 

Three independent reviewers recommended accepting your manuscript after some revisions. Please see the comments and address the points.

Sincerely,

jong-Yil Chai

Associate Editor

Mar Siles-Lucas

Deputy Editor

Three independent reviewers recommended accepting your manuscript after some revisions. Please see the comments and address the points.

Reviewer's Responses to Questions

**Key Review Criteria Required for Acceptance?**

**Methods**

-Are the objectives of the study clearly articulated with a clear testable hypothesis stated?

-Is the study design appropriate to address the stated objectives?

-Is the population clearly described and appropriate for the hypothesis being tested?

-Is the sample size sufficient to ensure adequate power to address the hypothesis being tested?

-Were correct statistical analysis used to support conclusions?

-Are there concerns about ethical or regulatory requirements being met?

Reviewer #1: Overall, I think the methods are reasonable. The only major concern I have is that when compared with Ref 29, are there any improvements in the methods, or the methods used here are exactly the same as in Ref 29 and the only difference is the input data?

Minor comments:

L80: "standardized prevalence rate" - how was it standardized?

L100: Why the infection rate (6.37%) is different from that presented on L98 (2.79%)?

L105-108: Is this true for the whole China too?

L117: Since Ref 29 was also conducted by the same research group, I suspect that you also have access to the nationwide data? Why only map Guangdong data in the current study?

L157: Could you be more specific about "according to the geographical location or natural environment" (i.e., east, middle, west? or plain and mountains?) and how many sectors there were?

L160: "thirdly, several sample units..." - How many sample units and how to determine the number of sample units for each county?

L161: "All samplings were randomized". - What do you mean here?

L175: For the land cover types, what about Grassland, village, and bare land? Are they included in "unclassified"?

Table1: The footnotes could be included as a column in the table.

L198: What do you mean by "all categorical variables are dumbed"?

L206: "Bernoulli distribution" - I think you mean "binomial distribution"? The outcome of a Bernoulli distribution is binary.

L209: "age groups k (k = 1, 2, 3, 4, 5)" - What are the age ranges for each group?

L227: "less informative priors" - LESS compared with what? I think you mean weak priors or noninformative priors?

L233: Why not use continuous variables directly in the model?

L255: ps could also be written as 1 - (1-p_A)*(1 - p_r)*(1-p_h), which is easier to understand?

Reviewer #2: (No Response)

Reviewer #3: The study’s methodology was well built-up. 

The objectives of the study was clearly stated with testable hypothesis.

Statistical analysis was correctly applied to support hypotheses and conclusions.

The surveys of were approved by the ethics committee, and a written consent form was obtained after oral explanation of objectives, procedures, and potential risks.

**Results**

-Does the analysis presented match the analysis plan?

-Are the results clearly and completely presented?

-Are the figures (Tables, Images) of sufficient quality for clarity?

Reviewer #1: L265: Maybe remove the commas in the number of individuals since commas are used both in the numbers and between numbers, which is confusing, or you could just remove this sentence, since the numbers are presented in table 2.

Figs S2-10: Putting the age groups as the panel title could be easier for the readers to get the differences between panels

S11 Text: Are these results based on the training or validating samples?

Reviewer #2: (No Response)

Reviewer #3: The result presented clearly following the intended plan.

The risk maps appropriately denote the regions with higher STH prevalence with calculated standard deviation.

**Conclusions**

-Are the conclusions supported by the data presented?

-Are the limitations of analysis clearly described?

-Do the authors discuss how these data can be helpful to advance our understanding of the topic under study?

-Is public health relevance addressed?

Reviewer #1: (No Response)

Reviewer #2: (No Response)

Reviewer #3: The conclusions drawn by the survey seem to play an essential role in implementing control programs. 

The limitations of the study are clearly described.

**Editorial and Data Presentation Modifications?**

Reviewer #1: (No Response)

Reviewer #2: (No Response)

Reviewer #3: Please modify the minor points in your manuscript as mentioned below.

Lines 105 – ‘spatial targeted’ to ‘spatially targeted’ or ‘spatial-targeted.’

Lines 228 – It might be better to explain why the authors applied less informative prior distributions for Bayesian models.

Lines 398 – ‘uses’ to ‘used.’

**Summary and General Comments**

Reviewer #1: This manuscript use environmental and socio-economic factors to interpolate the point STHs prevalence survey data in Guangzhou to a raster layer. The methods and results are well-presented and easy to follow. Although the results might be valuable for understanding the spatial distribution and its changes across time in Guangdong, there is not much novelty in the methods if it is exactly the same as in Ref 29.

Reviewer #2: (No Response)

Reviewer #3: This original article is about “The spatial-temporal distribution of soil-transmitted helminth infections in Guangdong Province, China: a geostatistical analysis of data derived from the three national parasitic surveys.” The paper is well written in describing the distribution of STH infections in Guandong Province. The study’s methodology was well built-up, and the map drawn by the survey seems to play an essential role in implementing control programs. 

There are some suggestions for authors,

1. If the survey data are not available, sharing R-codes with the public by data repositories like GitHub will be a good alternative.

2. How about presenting the ‘probability prevalence exceeds threshold map’? The threshold can impose the overall prevalence of China or the Province, indicating the regions require control programs in priority.

PLOS authors have the option to publish the peer review history of their article (what does this mean?). If published, this will include your full peer review and any attached files.

Reviewer #1: No

Reviewer #2: No

Reviewer #3: No

Figure Files:

Data Requirements:

Reproducibility:

References

---

## [Decision Letter · Decision Letter 1]

29 Jun 2022

Dear Dr. Lai,

We are pleased to inform you that your manuscript 'The spatial-temporal distribution of soil-transmitted helminth infections in Guangdong Province, China: a geostatistical analysis of data derived from the three national parasitic surveys' has been provisionally accepted for publication in PLOS Neglected Tropical Diseases.

Best regards,

Jong-Yil Chai

Associate Editor

Mar Siles-Lucas

Deputy Editor

Your revised manuscript has been reviewed by three reviewers. All of them recommended accepting it, and I concur. Thank you for your kind cooperation.

Reviewer's Responses to Questions

**Key Review Criteria Required for Acceptance?**

**Methods**

-Are the objectives of the study clearly articulated with a clear testable hypothesis stated?

-Is the study design appropriate to address the stated objectives?

-Is the population clearly described and appropriate for the hypothesis being tested?

-Is the sample size sufficient to ensure adequate power to address the hypothesis being tested?

-Were correct statistical analysis used to support conclusions?

-Are there concerns about ethical or regulatory requirements being met?

Reviewer #1: (No Response)

Reviewer #2: (No Response)

Reviewer #3: The study design was appropriate to describe the spatio-temporal distribution of STH infections in Guangdong

Province.

Explanatory analyses clearly articulated the target population, and the population was approprate for the hypothesis being tested.

The study included a total of 90931 individuals and 205 survey areas. The sample size seems to be sufficient to address the hypothesis being tested.

The authors used Bayesian method to estimate posterior inference, and handle uncertainty of the data, which seems appropriate in the setting of the study.

**Results**

-Does the analysis presented match the analysis plan?

-Are the results clearly and completely presented?

-Are the figures (Tables, Images) of sufficient quality for clarity?

Reviewer #1: (No Response)

Reviewer #2: (No Response)

Reviewer #3: The authors showed the results of bayesian variable selection, which are consistent with previous studies with similar environmental settings.

The risk maps of estimated infection prevalence clearly presents the regions which require control programs in priority.

**Conclusions**

-Are the conclusions supported by the data presented?

-Are the limitations of analysis clearly described?

-Do the authors discuss how these data can be helpful to advance our understanding of the topic under study?

-Is public health relevance addressed?

Reviewer #1: (No Response)

Reviewer #2: (No Response)

Reviewer #3: The conclusion is appropriate that the study can provide valuable information to assist control and prevention strategies for the province.

**Editorial and Data Presentation Modifications?**

Reviewer #1: (No Response)

Reviewer #2: (No Response)

Reviewer #3: (No Response)

**Summary and General Comments**

Reviewer #1: (No Response)

Reviewer #2: (No Response)

Reviewer #3: (No Response)

PLOS authors have the option to publish the peer review history of their article (what does this mean?). If published, this will include your full peer review and any attached files.

Reviewer #1: No

Reviewer #2: No

Reviewer #3: No

---

## [Editor Report · Acceptance letter]

14 Jul 2022

Dear Dr. Lai,

We are delighted to inform you that your manuscript, "The spatial-temporal distribution of soil-transmitted helminth infections in Guangdong Province, China: a geostatistical analysis of data derived from the three national parasitic surveys," has been formally accepted for publication in PLOS Neglected Tropical Diseases.

Best regards,

Shaden Kamhawi

co-Editor-in-Chief

Paul Brindley

co-Editor-in-Chief
